# Designing an Interpretability-Based Model to Explain the Artificial Intelligence Algorithms in Healthcare

**DOI:** 10.3390/diagnostics12071557

**Published:** 2022-06-26

**Authors:** Mohammad Ennab, Hamid Mcheick

**Affiliations:** Department of Computer Sciences and Mathematics, University of Québec at Chicoutimi, Chicoutimi, QC G7H 2B1, Canada; hamid_mcheick@uqac.ca

**Keywords:** interpretability, artificial intelligence, relative weights, probability

## Abstract

The lack of interpretability in artificial intelligence models (i.e., deep learning, machine learning, and rules-based) is an obstacle to their widespread adoption in the healthcare domain. The absence of understandability and transparency frequently leads to (i) inadequate accountability and (ii) a consequent reduction in the quality of the predictive results of the models. On the other hand, the existence of interpretability in the predictions of AI models will facilitate the understanding and trust of the clinicians in these complex models. The data protection regulations worldwide emphasize the relevance of the plausibility and verifiability of AI models’ predictions. In response and to take a role in tackling this challenge, we designed the interpretability-based model with algorithms that achieve human-like reasoning abilities through statistical analysis of the datasets by calculating the relative weights of the variables of the features from the medical images and the patient symptoms. The relative weights represented the importance of the variables in predictive decision-making. In addition, the relative weights were used to find the positive and negative probabilities of having the disease, which indicated high fidelity explanations. Hence, the primary goal of our model is to shed light and give insights into the prediction process of the models, as well as to explain how the model predictions have resulted. Consequently, our model contributes by demonstrating accuracy. Furthermore, two experiments on COVID-19 datasets demonstrated the effectiveness and interpretability of the new model.

## 1. Introduction

In recent years, with the development of artificial intelligence (AI), machine learning (ML), and deep learning (DL), technologies have achieved great success in computer vision, natural language processing, speech recognition, and other fields. ML models have also been widely applied to vital real-world tasks, such as face recognition [1,2], malware detection, and smart medical analysis [3]. Although AI outperforms humans in many meaningful tasks, its performance and applications have also been questioned due to the lack of interpretability [4]. The model is interpretable because it is small and basic enough to be completely comprehended. Ideally, the user should understand the learning process well enough to realize how it forms the decision limits from the training data and why the model has these rules [5]. For ordinary users, the machine-learning model, particularly the deep neural networks (DNN) model, is similar to a black box. We give it an input, and it feeds back a decision result. No one can know exactly the decision basis behind it and whether the decisions it makes are reliable. With the wide applications of AI solutions in healthcare, it becomes increasingly critical to improve the understanding of the working mechanism of the model and publish the white box of artificial intelligence [6]. Building trust in the machine-learning models has become a prerequisite for the ultimate adoption of AI systems. Hence it is crucial to improve model transparency and interpretability, specifically in high-risk areas that require reliability and security, e.g., healthcare. Due to an incomplete understanding of the working mechanism of the model, the model may produce results inconsistent with medical institutions, resulting in negative consequences in specific cultures and contexts. Based on the interpretable method, it is possible to clarify how the model makes each decision. Hence, each output result can be traced back, and the model results are more controllable [7]. To improve the interpretability and transparency of the machine-learning models, there is a need to establish a trusting relationship between users and decision-making models in practical deployment applications [8]. Therefore, to contribute to tackling the interpretability challenge, we designed a new interpretability model to explain the reasoning behind the prediction of the AI models in the healthcare domain. The basic principle of the new model is based on the statistics and probability rules by finding the relative weights of the variables which represent their relative importance in determining the prediction and the probability of having the disease. The positive likelihood ratio (+*LR*) indicates how likely it is that the patient has COVID-19, given a positive test result [9]. Similarly, the negative likelihood ratio (−*LR*) indicates how likely it is that the patient does not have COVID-19, given a negative test result [9]. The variables are either the symptoms of the patient or the characteristics of the affected parts of the organ as shown in a medical image. Calculating the relative weights is performed by dividing the weight of each variable by the sum of all weights of the variables in the dataset. Subsequently, the probability of positive infection is calculated by adding the related relative weights of the (i) characteristics of the affected parts of the organ as shown in a medical image or (ii) the symptoms of the patient. Our model contains two interpretability algorithms. Specifically, our model shows an alternative solution of deep learning and rules-based algorithms based on the relative weights of the variables of the healthcare diseases.

The rest of this paper is organized as follows. In Section 2, we review the background techniques that are rooted in a strong theoretical foundation. Section 3 explores the state of the art in addition to the related works and their limitations. In Section 4, we design the algorithms that create the explanations. In Section 5, the new model is validated by a real dataset. Section 6 discusses the solutions that our model provides. Finally, the conclusion and recommendations appear in Section 7.

## 2. Motivation

Healthcare presents particular ethical, legal, and regulatory problems since decisions can have an immediate impact on people’s well-being or lives [10]. One of the major implementation challenges highlighted is the inability to explain the decision-making progress of AI systems to physicians and patients [10]. Clinicians must be confident that AI systems can be trusted because they must provide the best treatment to each patient. Therefore, developing interpretable models might be a step toward trustworthy AI in healthcare. The area of explainable AI seeks to gain knowledge into how and why AI models make predictions while retaining high levels of predictive performance. Although the interpretability of AI models holds significant promises for health care, it is still in its early stages. Among other things, it is unclear what constitutes a sufficient explanation and how its quality should be assessed. Furthermore, the benefit of interpretability of AI systems has yet to be demonstrated in reality [11].

In this paper, we contribute to the larger goal of creating trustworthy AI models in healthcare, by designing a new model that is added to the state of the art of interpretability techniques as a contribution that implements statistics and probability rules to produce accurate interpretations for the predictions of the AI algorithms.

## 3. Background: Statistics and Probability Techniques

In general, the interpretability-based model is based on statistics and probabilities rules to train the datasets.

We distinguish two interpretability strategies that adopt probabilities in their findings with solid theoretical backgrounds and are easy to implement: the Locally Interpretable Model-Agnostic Explanations (LIME), and (ii) the *Deep Learning Important FeaTures*.

### 3.1. Locally Interpretable Model-Agnostic Explanations (LIME)

Ribeiro et al. [12] introduced a surrogate model that uses a trained local model to interpret a single sample. However, the black-box model is explained by taking an instance sample of interest, performing disturbance near it to generate new sample points, and obtaining its predicted value. LIME uses the new dataset to train an interpretable model (such as linear regression or a decision tree) to obtain a near-local approximation to the black-box model. LIME consists of two parts, LIME, and SP-LIME, while LIME approximates the model with a fidelity method, SP-LIME is used to select non-redundant instances (basically covering all features) to explain the global behavior of the model. Additionally, LIME can interpret the classification results of the medical image and can also be applied to related tasks of natural language processing, such as topic classification, part-of-speech tagging, etc. Because the starting point of LIME itself is model-independent, it has broad applicability [13].

### 3.2. Deep Learning Important FeaTures or DeepLIFT

DeepLIFT is a method for dissecting the output prediction of the neural network on a given input by backpropagating the contributions of all neurons in the network to each characteristic of the input. DeepLIFT assigns value to neurons depending on their activity. When the local gradient is zero, the findings might be deceptive. DeepLIFT produces surprisingly distinct attribution maps from input CT images with minor perturbations that are visually identical. Moreover, DeepLIFT can also show dependencies that other methods overlook by distinguishing between the negative and positive contributions [14].

### 3.3. Definition of Concepts

Some neighboring concepts are occasionally used as synonyms for transparency, including interpretability, explainability, and understandability [15]. However, there is a subtle difference between explainability and interpretability. The model is considered inherently interpretable if a person can comprehend its underlying workings, either the complete model at once or at least the elements of the model relevant to a specific prediction. It may include understanding decision criteria and cut-offs and the ability to compute the model’s outputs manually. In contrast, we consider the model’s prediction explainable if a process can offer (partial) knowledge about the model’s workings. These details include identifying which elements of input were most significant for the resulting forecast or which adjustments to input would result in a different prediction [5]. Moreover, transparency implies that the behavior of artificial intelligence and its related components are understandable, explainable, and interpretable by humans. Besides, understandability means that the decisions made by the artificial intelligence model can reach a certain degree of understanding [16].

Before introducing specific interpretability problems and corresponding solutions, we briefly introduce what interpretability is and why it is needed. Data mining and machine-learning scenarios define interpretability as the ability to explain and present understandable terms to humans [17]. In essence, interpretability is the interface between humans and the decision model, which is both an accurate proxy for the decision model and understandable by humans [18]. In top-down machine learning, which builds models on a set of statistical rules and assumptions, interpretability is critical because it is the cornerstone of the defined rules and assumptions. Furthermore, model interpretability is a critical means of verifying that the assumptions are robust and that the defined rules are well suited to the task. Unlike top-down tasks, bottom-up machine learning usually corresponds to the automation of manual and onerous tasks. Given a batch of training data, the model automatically learns the input data and output categories by minimizing the learning error in the mapping relationship between them. In bottom-up learning tasks, the model is built automatically, so we do not know its learning process or its working mechanism. Therefore, interpretability aims to help people understand how a machine-learning model learns [8].

## 4. State of the Art

### 4.1. Related Works

In recent years (2016–2022), the interpretability of the various artificial intelligence models has attracted great attention from the academic and business sectors. Researchers have successively proposed several interpretation methods to solve and enhance the model “black box” problem. We distinguished three interpretability categories that have characteristics, advantages, and disadvantages. Rules-based interpretation models: A linear method was proposed in [19] by adding regularization to the tree, reducing the nodes of the decision tree, and solving the problem of a vast number of nodes without losing accuracy. An interpretable tree framework was proposed in [20] which can be applied to classification and regression problems by extracting, measuring, pruning, and selecting rules from tree collections and computing frequent variable interactions. This model also forms a rules-based learner, a reduced tree ensemble learner (STEL), and uses it in prediction. A method was proposed in [21] that learns rules to globally explain the behavior of black-box machine-learning models used to solve classification problems. It works by first extracting the important conditions at the instance level and then going through a genetic algorithm with suitable fitness-function rules. These rules represent the patterns in which the model makes decisions and help to understand its behavior.

### 4.2. Bayesian Nonparametric Approach

Guo et al. [11] designed a Bayesian nonparametric model to define an infinite-dimensional parameter space. In other words, the size of this model can adapt to the change in the AI model as the data are increased or decreased. This model can be determined according to how many data parameters are selected. It only needs a small assumption to learn data and perform clustering. The increasing data also can be continuously aggregated into corresponding classes. At the same time, this model also performs predictions. According to the specific learning problem, a spatial data model composed of all parameters related to this problem can be solved.

#### 4.2.1. GAM

A global variable generalized additive weight method called GAM was proposed in [4], which accounts for the pattern of neural-network predictions of swarms. The global interpretation of GAM describes the nonlinear representation learned by the neural network. GAM also provides adjustable subpopulation granularity and the ability to track global interpretations for specific samples.

#### 4.2.2. MAPLE

MAPLE may be used nearly identically as an explanation for a black-box model or as a predictive model; the main difference is that in the first instance MAPLE is fitted to the black-box model’s prediction, whilst in the second situation MAPLE is fitted to the response variable. MAPLE has various intriguing feature LIMEs, which are mentioned below: (i) It avoids the trade-off between model performance and model interpretability since MAPLE is a highly accurate predictive model capable of generating correct predictions. (ii) It finds global trends by using local examples and explanations. MAPLE stands out from other interpretability frameworks due to its training distributions [22].

#### 4.2.3. Anchors

Anchors is a model-independent, rule-based local explainer approach [23]. Anchors ensure that the forecasts of occurrences in the same anchor are nearly identical. In other words, anchors identify the qualities that are sufficient to correct the forecast while modifying the other attributes that do not affect the prediction. The bottom-up approach, in which anchors are built sequentially, is one method of anchor construction. Anchors, in particular, begin with an empty rule and extend it with one feature in each iteration until the resulting rule has the greatest estimated accuracy [23].

#### 4.2.4. SHAP

A game theory concept was used to quantify the impact of each feature on the prediction process. The Shapley value [24] is a mechanism from coalitional game theory that can properly distribute benefits across players (features) when players’ contributions are uneven. In other words, Shapley values are founded on the assumption that characteristics work together to influence the model’s prediction toward a specific value. It then attempts to distribute its contributions fairly across all featured subsets. Shapley value, in particular, distributes the difference between prediction and average prediction equitably among the feature values of the instance to be explained. Shapely value fulfills three intriguing features [25].

#### 4.2.5. Perturbation-Based Methods

Perturbation is the most basic method for examining the impact of modifying an AI model’s input properties on its output. This can be accomplished by eliminating, masking, or changing specific input variables, then conducting the forward pass (output calculation) and comparing the results to the original output. This is comparable to the sensitivity analysis conducted in parametric control system models. The input features that have the greatest influence on the output are ranked first. It is computationally intensive since a forward pass must be performed after perturbing each collection of characteristics in the input [26]. In the case of picture data, the perturbation is accomplished by covering sections of the image with a grey patch and thereby obscuring them from the system’s perspective. It can give both positive and negative evidence by identifying the responsible characteristics [27].

#### 4.2.6. Attention Based

The fundamental concept of attention is motivated by the way people pay attention to various areas of a picture or other data sources in order to interpret them. The technique employed attention mechanisms to show the detection process, which included an image model and a language model [28]. The language model discovered dominant and selective characteristics to learn the mapping between visuals and diagnostic reports using that attention mechanism [26].

#### 4.2.7. Concept Vectors

A unique approach was developed in [29] called Testing Concept Activation Vectors (TCAV) to explain the characteristics learnt by successive layers to domain experts who do not have deep-learning knowledge in terms of human-understandable concepts. It used the directional derivative of the network in idea space, similar to how saliency maps use it in input feature space. It was put to the test to explain DR level predictions, and it effectively recognized the existence of microaneurysms and aneurysms in the retina. This gave medical practitioners with explanations that were easily interpretable in terms of the existence or absence of a specific notion or physical structure in the image. [26]. Many clinical notions, such as the texture or form of a structure, cannot be adequately defined in terms of presence or absence and require a continuous scale of assessment.

#### 4.2.8. Similar Images

In [30], research was proposed analyzing layers of a 3D-CNN using a Gaussian mixture model (GMM) and binary encoding of training and test pictures based on their GMM components to yield comparable 3D images as explanations. As an explanation for its conclusion, the algorithm returned activation-wise similar training pictures utilizing atlas. It was proven on 3D MNIST and an MRI dataset, where it yielded pictures with identical atrophy conditions. However, it was shown that in some circumstances, activation similarity was dependent on the spatial orientation of pictures, which might influence the choice of the returned images [26].

#### 4.2.9. Textual Justification

This model explained its decision in terms of words or phrases that describe its logic and may communicate directly with both expert and ordinary users [26]. A justification model that utilized inputs from the classifier’s visual characteristics, as well as prediction embeddings, was used to construct a diagnostic phrase and visual heatmaps for breast-mass categorization [31]. In order to develop reasons in the presence of a restricted quantity of medical reports, the justification generator was trained using a visual word constraint loss [26].

#### 4.2.10. Intrinsic Explainability

Intrinsic explainability explains its decisions in terms of human visible decision limits or variables. For a few dimensions where the decision boundaries can be viewed, they generally comprise relatively simpler models such as regression, decision trees, and SVM [32].

#### 4.2.11. Recurrent Neural Network (RNN)

Ref. [33] proposed RNN model that combined a two-layer attention mechanism for sequential numbers according to the data. The method gave a detailed explanation of the prediction results and retained the relative accuracy of RNN.

### 4.3. Limits of the Existing Solutions

Significant progress has been made in explaining the decisions of the AI models, particularly those implemented in medical diagnosis. Understanding the features responsible for a particular prediction helps model designers iron out dependability problems so that end users may acquire trust and make better decisions [26]. Almost all of these strategies aim towards local explainability or justifying decisions in a particular case. However, it is essential to consider the characteristics of a black-box that might make the wrong decision for the wrong reason. It is a significant issue that can have an impact on performance when the system is implemented in the real world [26].

Moreover, when considering the above interpretability methods, there is a lack of quantitative judgments, which indicates their low explanation fidelity. The AI algorithms in the healthcare domain make their decisions in the domain of positive or negative test results, which explains their low accuracy [34]. However, there is a need to make explainability approaches more comprehensive and intertwined with uncertainty methods [26]. Moreover, it is essential to consider the characteristics of a black-box that might make the wrong decision for the wrong reason as it is a significant issue that can have an impact on the performance when the system is implemented in the real world [26]. In response and to take a role in tackling these challenges, we contribute to the field by designing an interpretability-based model which explains the predictions of the AI algorithms in healthcare. This approach simulates human-like reasoning abilities and makes the explanations by describing the various features of the medical image and the symptoms of the patient. We validate our model by performing experiments on COVID-19 datasets to demonstrate its effectiveness and interpretability. Table 1 shows the limitations of the existing interpretability methods.

## 5. The Interpretability-Based Model

The core principle of the new model is the using of the statistics and probability rules to train the datasets, by finding the relative weights of the variables which represent their relative importance in determining the prediction and the probability of having the disease. The variables are either the symptoms of the patient or the characteristics of the affected parts of the organ as shown in the medical image. Calculating the relative weights is performed by dividing the weight of each variable by the sum of all weights for all variables. Subsequently, the result of training the dataset is the likelihoods of positive infection. Our model contains two interpretability algorithms for training the datasets. The first is to interpret the predictions of the neural-networks models, shown in Figure 1. The second explains the decisions of the rules-based models more precisely, as shown in Figure 2. The new model generates the predictions and their predictive values by executing three steps as shown in Figure 3 (i) inputting the dataset: the classification features of the medical image dataset or the symptoms of the patient dataset; (ii) data preprocessing: defining the variables which include the symptoms and the characteristics of the classification features for the medical image; (iii) finding the relative weight of each variable using the formula:$$Relative\;weight=\frac{Weight\;of\;the\;variable}{\sum Weights\;of\;all\;variables}$$

(iv) finding the probabilities: calculating the negative and positive probabilities for each explanation using the formulas:+*LR* = ∑*Relative weights of the variables*
−*LR* = 1 − (+*LR*)

**Figure 1 diagnostics-12-01557-f001:**
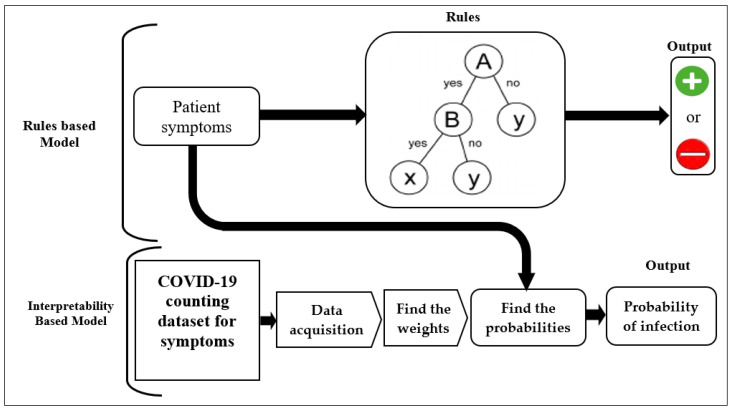
Using the interpretability-based model to interpret the predictions of the rules-based model for COVID-19 patients given their symptoms.

**Figure 2 diagnostics-12-01557-f002:**
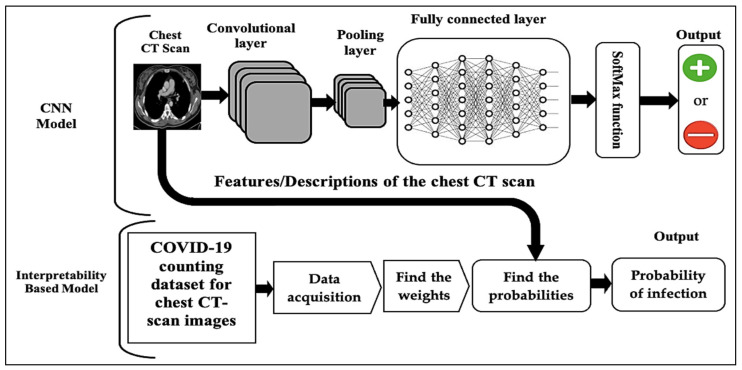
Using the interpretability-based model to interpret the predictions of the CNN model for COVID-19 patients given their chest CT images.

**Figure 3 diagnostics-12-01557-f003:**
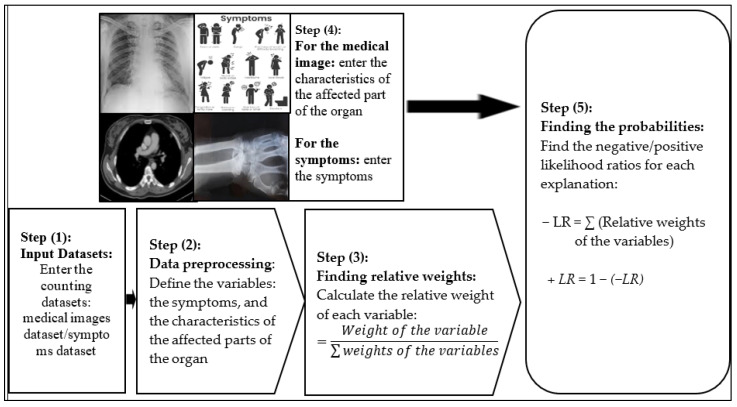
Using the interpretability-based model to train the datasets.

### 5.1. Dataset Requirements

The datasets must be labeled and representative. They must include all the salient features of the disease’s medical image and the symptoms. Furthermore, they should include the counting details, i.e., the number of variables with specific descriptions, characteristics, and values.

### 5.2. Defining the Variables

The variables of the interpretable-based model are (i) the symptoms of the disease and (ii) the spatial features of the medical image.

### 5.3. Relative Weights as Ranking

Explanations contain variables encoded by treating each as a vector in *R^n^*: the relative weight ranks and considers each variable’s association with a particular prediction. Each variable is a vector that answers the question: How important is this variable for a particular prediction? Or what is the extent for this variable to be in the decision of the machine-learning model? We treat variables as relative weighted conjoined rankings.

### 5.4. Creating the Explanations

The interpretability-based model algorithms use the relative weights of the variables, to define the average of repetitions of the variables’ characteristics in the dataset, which measures the relative importance of the variables in the prediction of the AI algorithm.

The outcomes of our model include the positive and negative probabilities of having the disease.

## 6. Validating the Interpretability-Based Model

Research in interpretability inherently faces the challenge of effective and reliable validation [17]. Identifying the appropriate validation methodology for a proposed approach is an open research question [35]. This paper validates the interpretability-based model using real datasets for COVID-19 patients. We use one dataset for medical images and another for symptoms as shown in Table 2 and Table 3, respectively.

The data were gathered from open access data [36] and evaluated to improve clinical decisions and treatment. There is a total of 112 confirmed COVID-19 patients (range, 12–89 years), including 51 males (range, 25–89 years) and 61 females (range, 12–86 years). The data is public and available on the web [36].

### 6.1. Validating Our Model to Interpret the Predictions of the Neural-Networks Model

The neural-networks model is used to provide the predictions for COVID-19 patients by training the dataset of the chest CT images and making the prediction based on analyzing the medical image of the tested patient. However, the machine-learning models do not produce explanations for the predictions.

The suggested model will provide the interpretation based on the saved counting data for a set of chest CT images that are shown in Table 2 and Figure 4, where the relative weights are calculated by dividing the weight of each variable by the sum of all weights.

We validate the model by applying Algorithm 1 of the interpretability-based model to the counting data in Table 2, as shown in Figure 2:(1)Define the variables of the explanation set: distribution of pulmonary lesions (no lesion, peripheral, central, diffuse), involvement of the lung (no involvement, single lobe, unilateral multilobe, bilateral multilobe), GGO, crazy-paving pattern, consolidation, linear opacities, air bronchogram, cavitation, bronchiectasis, pleural effusion, pericardial effusion, lymphadenopathy, and pneumothorax.(2)Train the dataset by finding the relative weights of the variables as shown in Table 4 and generating all the probable explanations for the patient by finding the sum of the related relative weights and calculating the positive and negative probabilities using the following formulas:
−*LR* + = *relative weights of the variables*
+*LR* = 1 − (−*LR*)

**Table 4 diagnostics-12-01557-t004:** The variables of the chest CT image, along with their calculated relative weights.

Variable or Feature	Relative Weight (=$\frac{The\;weight\;of\;the\;variable}{Total\;number\;of\;weights\;for\;all\;variables})$
**Distribution of pulmonary lesions**	
No lesion	3.2%
Peripheral	13.1%
Central	0.3%
Diffuse	5.2%
**Involvement of the lung**	
No involvement	3.2%
Single lobe	5%
Unilateral multilobe	0.4%
Bilateral multilobe	13.5%
**GGO**	18.5%
**Crazy-paving pattern**	10%
**Consolidation**	9.4%
**Linear opacities**	7.3%
**Air bronchogram**	6.3%
**Cavitation**	0%
**Bronchiectasis**	4.4%
**Pleural effusion**	2.8%
**Pericardial effusion**	0.3%
**Lymphadenopathy**	0%
**Pneumothorax**	0.3%

For instance, if the physician suspects a COVID-19 case by applying a deep learning model on the CT image, they may use the interpretability-based model to explain the prediction for the patient. In order to do that, they should observe the existing features of the CT image e.g., GGO, bronchiectasis, pericardial effusion, consolidation, involvement of the lung (bilateral multilobe), distribution of pulmonary lesions (peripheral). Using our model, the positive likelihood ratio (+*LR*) according to Table 4 is:18.5% + 4.4% + 0.3% + 9.4% + 13.5% + 13.1% = 59.2%

Whereas the negative likelihood ratio (−*LR*) is:1 − 59.2% = 40.8%

The physician can explain the likelihood ratios for the patient using the indication of the relative weights of the symptoms. Table 5 includes all the possible predictions of our model based on the trained dataset. Where (+) represents the existence of the symptom and (−) represents its absence.

### 6.2. Validating Our Model to Interpret the Predictions of the Rules-Based Models

Generally, the rules-based models provide low explanation fidelity because their decisions are in the domain of positive or negative results. However, we apply the interpretability rules-based Algorithm 2 to the dataset in Table 3 that includes the symptoms of COVID-19 patients who have at most 13 symptoms to generate the explanations as shown in Figure 1 and as in the following:1.Define the variables of the explanation model which will be the symptoms: Fever, Dizziness, Palpitation, Throat pain, Nausea and vomiting, Headache, Abdominal pain and diarrhea, Expectoration, Dyspnea, Myalgia, Chest distress, Fatigue, and Dry Cough.2.Train the dataset by calculating the relative weights for the variables by dividing the ratio weight of each symptom by the sum of all weights, as shown in Table 6, and generate the explanations for the patient by finding the sum of the related relative weights and the positive and negative probabilities:3.−*LR* + = *relative weights of the variables*4.+*LR* = 1 − (−*LR*)

**Table 6 diagnostics-12-01557-t006:** The variables of the chest CT image, along with their calculated relative weights.

Variable or Symptom	Relative Weight (=$\frac{The\;weight\;of\;the\;variable}{Total\;number\;of\;weights\;for\;all\;variables})$
Dizziness	0.7%
Palpitation	0.7%
Throat pain	1.4%
Nausea and vomiting	1.4%
High-grade fever (>39.0)	2.5%
Headache	2.9%
Abdominal pain and diarrhea	5%
Expectoration	5.4%
Dyspnea	6.5%
Myalgia	6.5%
Chest distress	8.6%
Moderate-grade fever (38.1–39.0)	11.1%
Fatigue	13.6%
Low-grade fever (37.3–38.0)	16.5%
Dry Cough	17.2%


**Algorithm 1:** Interpretability algorithm for training the dataset of the neural-network models1. Input: the characteristics of the affected parts of the organ as per the medical image2. Variables = the set of the characteristics of the affected parts of the organ/*3. For each variable assign a relative weight*/

$Relative\;weight=\frac{Weight\;of\;the\;variable}{\sum Weights\;of\;all\;variables}$

/*4. Generate the probabilities of having the disease*/            −*LR* + = *relative weights of the variables*                                +*LR* = 1 − (−*LR*)5. Output: the positive and negative probabilities in addition to the relative weights of the variableEnd



**Algorithm 2:** Interpretability algorithm for training the dataset of the rules-based models1. Input: the symptoms of the patient variable2. Variables = the set of symptoms/*3. For each symptom assign a relative weight*/ 

$Relative\;weight=\frac{Weight\;of\;the\;variable}{\sum Weights\;of\;all\;variables}$

/*4. Generate the probabilities of having the disease*/            −*LR* + = *relative weights of the variables*
                                +*LR* = 1 − (−*LR*)5. Output: the positive and negative probabilities in addition to the relative weights of the variableEnd


For example, if a patient exhibits signs and symptoms of COVID-19 e.g., dry cough, fever (37.9 °C), headache, and myalgia, the physician suspects a COVID-19 case and recommends the patient take a polymerase chain reaction (PCR) test. The physician may use the interpretability-based model to explain the prediction for the patient. According to our model, the positive likelihood ratio (+*LR*) based on Table 6 is:17.2% + 16.5% + 2.9% + 17.2% + 6.5% = 60.3%

Whereas the negative likelihood ratio (−*LR*):1 – 60.3% = 39.7%

The physician can explain the likelihood ratios for the patient using the indication of the relative weights of the symptoms. Table 7 includes all the possible predictions of our model based on the trained dataset. Where (+) represents the existence of the symptom and (−) represents its absence.

## 7. Discussion

Our model represents a significant progress in explaining the decisions of the AI models in medical diagnosis. Additionally, understanding the features responsible for a particular prediction helps model designers iron out dependability problems so that end users may acquire trust and make better decisions [26]. The interpretability-based model contributes in: (i) reducing the cost of mistakes in the medical domain that is highly effected by the wrong prediction. (ii) minimizing the influence of AI model bias by elaborating on the decision-making criteria which builds trust for all users [37]. 

In this section, we highlight and compare our model with the other interpretability methods that were described in Section 4. When we apply the datasets of symptoms and CT images, we find that our model is able to distinguish the feature importance which is represented by the relative weight of the variable. In addition, it satisfies the model independent from other related algorithms as our model applies the statistics and probability rules. Moreover, the new model is able to identify the relative features for each instance. However, the accuracy of the predictions of the interpretability-based model is a limitation, because it depends on the correctness of the trained dataset, which may not exist in some medical images where the readings of the variables in the dataset are determined by the related experience of the medical staff. 

## 8. Conclusions and Future Works

In this paper, we build an interpretability-based model—a methodology for making explanations to supplement existing interpretability techniques for the AI algorithms based on the statistics and probability rules.

Our model provides the decisions and the explanations. Additionally, it provides the likelihood of positive and negative infections and the ability to trace explanations to specific samples. We demonstrated the use of our model using real datasets for COVID-19 patients, one for the CT image and another for the rules-based model for symptoms, where the suggested model illuminates the explanation patterns across learned sets. Furthermore, with explanations across subpopulations, convolutional neural network predictions are more transparent. A possible next step is to reduce the complexity of the interpretable rules-based algorithms that offset their interpretability. Another future work area is to apply our model on other industries to optimize their AI systems. e.g., marketing, insurance, financial services.

## Figures and Tables

**Figure 4 diagnostics-12-01557-f004:**
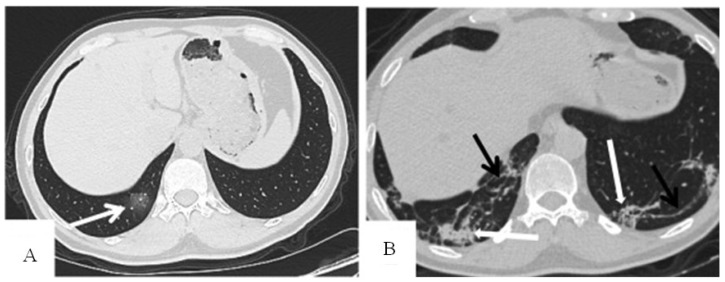
“Male, 48 years, transverse thin-section CT scan. (**A**) 4 days after the onset of initial symptoms: subpleural GGO (arrow) in the right lower lobe. (**B**) 38 days after the onset of initial symptoms: There was no significant change in the extent and composition of lesions compared with the directly prior CT result, with consolidation (thick white arrow) and linear opacities (thin black arrow) in the right lower lobe, and crazy-paving pattern (thick white arrow) and linear opacities (thin black arrow) in the left lower lobe” [36].

**Table 1 diagnostics-12-01557-t001:** The limitations of the existing interpretability methods.

Method	Feature Importance	Model-Independent	Individualized Feature Importance	Identifying the Set of Relevant Features for Each Instance
**LIME**	√		√	
**DeepLIFT**	√		√	
**SHAP**	√		√	
**Recurrent Neural Network (RNN)**	√			√
**MAPLE**	√		√	
**GAM**	√	√		√
**Rules-based interpretation models**	√			
**Anchors**	√			
**Textual justification**	√		√	
**Bayesian nonparametric approach**	√	√		√
**Intrinsic explainability**	√			
**Similar images**	√			√

**Table 2 diagnostics-12-01557-t002:** The counting data for the medical image used in the CNN model [36].

Variable (Feature)	Number of Patients (Min Weight of the Variable = $\frac{The\;number\;of\;patients\;who\;have\;the\;variable}{Total\;number\;of\;patients})$	Number of Patients (Max Weight of the Variable = $\frac{The\;number\;of\;patients\;who\;have\;the\;variable}{Total\;number\;of\;patients})$
**Distribution of pulmonary lesions**		
No lesion	1 (1.7%)	10 (21.2%)
Peripheral	31 (52.4%)	30 (63.8%)
Central	0 (0%)	1 (2.1%)
Diffuse	26 (44.1%)	6 (12.7%)
**Involvement of the lung**		
No involvement	1 (1.7%)	10 (21.2%)
Single lobe	1 (1.5%)	16 (34.0%)
Unilateral multilobe	0 (0%)	2 (2.9%)
Bilateral multilobe	65 (95.6%)	20 (42.5%)
**GGO**	52 (98.1%)	36 (76.5%)
**Crazy-paving pattern**	42 (62.7%)	17 (36.1%)
**Consolidation**	51 (75.0%)	12 (25.5%)
**Linear opacities**	49 (83.1%)	3 (6.3%)
**Air bronchogram**	27 (50.0%)	8 (17.0%)
**Cavitation**	0 (0%)	0 (0%)
**Bronchiectasis**	24 (45.2%)	3 (6.3%)
**Pleural effusion**	19 (27.9%)	2 (4.2%)
**Pericardial effusion**	3 (4.4%)	0 (0%)
**Lymphadenopathy**	0 (0%)	0 (0%)
**Pneumothorax**	2 (3.8%)	0 (0%)

**Table 3 diagnostics-12-01557-t003:** The counting dataset for COVID-19 patients who have at most 13 symptoms [36].

Variable or Symptom	No. of Patients	Relative Weight ($\frac{The\;weight\;of\;the\;variable}{The\;total\;of\;weights\;for\;all\;variables})$
Fever-low (37.3–38.0)	46	41.1%
Fever-moderate (38.1–39.0)	31	27.6%
Fever-high (>39.0)	7	6.2%
Dizziness	2	1.7%
Palpitation	2	1.7%
Nausea and vomiting	4	3.5%
Throat pain	4	3.5%
Headache	8	7.1%
Abdominal pain and diarrhea	14	12.5%
Expectoration	15	13.3%
Dyspnea	18	16.1%
Myalgia	18	16.1%
Chest distress	24	21.4%
Fatigue	38	33.9%
Dry Cough	48	42.8%

**Table 5 diagnostics-12-01557-t005:** All the probable predictions and explanations for COVID-19 patients according to their chest CT image.

Explanation	Distribution of Pulmonary Lesions				Involvement of the Lung																
Number	No Lesion	Peripheral	Central	Diffuse	No Involvement	Single Lobe	Unilateral Multilobe	Bilateral Multilobe	GGO	Crazy–Paving Pattern	Consolidation	Linear Opacities	Air Bronchogram	Cavitation	Bronchiectasis	Pleural Effusion	Pericardial Effusion	Lymphadenopathy	Pneumothorax	+*LR* (%)	−*LR* (%)
1	+	+	+	+	+	+	+	+	+	+	+	+	+	+	+	+	+	+	+	100	0
2	+	−	+	+	+	+	+	+	+	+	+	+	+	+	+	+	+	+	+	86.9	13.1
3	+	+	−	+	+	+	+	+	+	+	+	+	+	+	+	+	+	+	+	99.7	0.3
4	+	−	−	+	+	+	+	+	+	+	+	+	+	+	+	+	+	+	+	94.5	5.5
.	.	.	.	.	.	.	.	.	.	.	.	.	.	.	.	.	.	.	.	.	.
.	.	.	.	.	.	.	.	.	.	.	.	.	.	.	.	.	.	.	.	.	.
524,287	−	+	−	−	−	−	−	−	−	−	−	−	−	−	−	−	−	−	−	13.1	86.9
524,288	−	−	−	−	−	−	−		−	−	−	−	−	−	−	−	−	−	−	0	100

‘.’ Represents a probable prediction that is not mentioned.

**Table 7 diagnostics-12-01557-t007:** All the probable predictions and explanations for COVID-19 patients according to their symptoms.

Explanation Number	Fever (37.3–38.0)	Fever (38.1–39.0)	Fever (>39.0)	Dry Cough	Expectoration	Throat Pain	Chest Distress	Dyspnea	Fatigue	Nausea and Vomiting	Palpitation	Dizziness	Headache	Myalgia	Abdominal Pain and Diarrhea	+*LR* (%)	−*LR* (%)
1	+	+	+	+	+	+	+	+	+	+	+	+	+	+	+	100	0
2	+	-	+	+	+	+	+	+	+	+	+	+	+	+	+	88.9	11.1
3	+	+	-	+	+	+	+	+	+	+	+	+	+	+	+	97.5	2.5
4	+	-	-	+	+	+	+	+	+	+	+	+	+	+	+	86.4	13.6
5	+	+	+	-	+	+	+	+	+	+	+	+	+	+	+	82.8	17.2
6	+	-	+	-	+	+	+	+	+	+	+	+	+	+	+	71.7	28.3
.	.	.	.	.	.	.	.	.	.	.	.	.	.	.	.	.	.
.	.	.	.	.	.	.	.	.	.	.	.	.	.	.	.	.	.
.	.	.	.	.	.	.	.	.	.	.	.	.	.	.	.	.	.
.	.	.	.	.	.	.	.	.	.	.	.	.	.	.	.	.	.
8191	-	+	-	-	-	-	-	-	-	-	-	-	-	-	-	11.1	88.9
8192	-	-	-	-	-	-	-	-	-	-	-	-	-	-	-	0	100

‘.’ Represents a probable prediction that is not mentioned.

## Data Availability

We used datasets to validate the proposed model, these sets are published using the below link: https://www.ejradiology.com/article/S0720-048X(20)30198-4/fulltext (accessed on 17 February 2022).

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
