# Peer review of "Designing an Interpretability-Based Model to Explain the Artificial Intelligence Algorithms in Healthcare"

_diagnostics, 2022, doi:10.3390/diagnostics12071557_

Round 1

Reviewer 1 Report

In fact my assessment of the submission is somewhere between "requires major revisions" and 'reject".

The formal statement of the research problem described by the Authors is very poor - I can' agree that "the new model has outdone the available prediction models in demonstrating accuracy by providing not only the decisions but also their interpretation concurrently", as the Authors claim in the abstract.

The final results of the research, presented in Table 6, are based only on the symptoms which can be determined on the basis of a medical survey only! What is the role of the CT scans in the explanatory model? In fact the weights of the symptoms, presented in Table 5 (which are the basis of explanations) could be obtained from simple statistics of symptoms gathered from thousands of COVID-19 patients. Why (according to the Authors) was the algorithm of extraction the weights from deep NN better than such a simpel one?

In fact, the problem formulation, the tools used for its solving and the reasoning  algorithms are described in a way which does not make it possible to use the methodology by other researchers.

There are also many editorial errors, e.g. "(38.139.0)" in 13th and 15th row of Table 5, lack of reference number (line 135), lack of figuer number (the first paragraph of Section 4), etc., etc.

Author Response

Thank you very much for your valuable comments and feedback regarding our research paper. Your insight has served to strengthen our manuscript and we have made changes to reflect them. We have made the changes directly to the manuscript, as well as recorded the changes in the attached file.

Response Sheet

Manuscript ID: diagnostics-1721102

Title of the manuscript:  Designing Interpretability-based Model to Explain the Artificial
Intelligence Algorithms in Healthcare

 Response to reviewer 1 comments

Question: The formal statement of the research problem described by the Authors is very poor - I can' agree that "the new model has outdone the available prediction models in demonstrating accuracy by providing not only the decisions but also their interpretation concurrently", as the Authors claim in the abstract.

Answer: The abstract was updated accordingly

Question: The final results of the research, presented in Table 6, are based only on the symptoms which can be determined on the basis of a medical survey only! What is the role of the CT scans in the explanatory model? In fact the weights of the symptoms, presented in Table 5 (which are the basis of explanations) could be obtained from simple statistics of symptoms gathered from thousands of COVID-19 patients. Why (according to the Authors) was the algorithm of extraction the weights from deep NN better than such a simple one?

Answer: When validating the algorithms, we use the attributes for a couple of separate datasets: (i) the symptoms (ii) the characteristic of the affected parts of the organ as shown in the medical image. In a manner that using the relative weights and probabilities in explaining the predictions of the AI models is a contribution to the interpretability field where the high accuracy of the explanation is formulated through values. However, making the predictions based on COVID-19 symptoms without finding the weights may produce an approximate interpretation of having the disease which reflects inaccuracy. Similarly, for predicting the COVID-19 using the CT scan, finding the relative weights of the affected parts of the organ as shown on the medical image is the key to interpreting the probability of infection.

Question: In fact, the problem formulation, the tools used for its solving and the reasoning algorithms are described in a way which does not make it possible to use the methodology by other researchers.

Answer: The main objective of our paper is to design a new model that is added to the state of the art of interpretability techniques as a contribution that uses the statistical weights and probabilities concepts to produce accurate interpretations for the predictions of the AI algorithms. Additionally, A possible next step is to reduce the complexity of the interpretable rules-based algorithms that offsets their interpretability. Another future work is to investigate how global interpretability can aid in ensuring fairness in artificial intelligence algorithms.

Updates were placed for many sections in the article. Moreover, the motivation section was added.

Question: There are also many editorial errors, e.g. "(38.139.0)" in 13th and 15th row of Table 5, lack of reference number (line 135), lack of figuer number (the first paragraph of Section 4), etc., etc.

Answer: The referred errors were fixed.

Reviewer 2 Report

"Abstract: The lack of interpretability in artificial intelligence models (deep learning, machine learning, and linear-based) is an obstacle to their widespread adoption in healthcare." Isn't deep learning a type of the machine learning? Linear-based what?  There are lots of interpretable machine learning models, e.g. logical analysis of data (see https://www.mdpi.com/2073-8994/14/3/600 ), decision trees etc.

The introduction is senseless. The authors do not describe the main idea of their interpretable model or method. What is the difference between their contribution and known results?

In Section 2, the authors list only two approaches to building interpretable models with "solid theoretical backgrounds". However, modern literature contains much more approaches.

Thus, the bibliography overview is far from complete, and not systematized.

The proposed algorithms are given without needed details, the authors do not use mathematical notation usual for the works on AI.

The most important flaw of this work is that there is no comparison of the results with the known interpretable AI methods which can be used for the same medical problems.

Thus, it is impossible to estimate the importance of the results of this paper.

Author Response

Thank you very much for your valuable comments and feedback regarding our research paper. Your insight has served to strengthen our manuscript and we have made changes to reflect them. We have made the changes directly to the manuscript, as well as recorded the changes in the attached file.

Response Sheet

Manuscript ID: diagnostics-1721102

Title of the manuscript:  Designing Interpretability-based Model to Explain the Artificial
Intelligence Algorithms in Healthcare

 Response to reviewer 2 comments

Question: "Abstract: The lack of interpretability in artificial intelligence models (deep learning, machine learning, and linear-based) is an obstacle to their widespread adoption in healthcare." Isn't deep learning a type of the machine learning? Linear-based what?  There are lots of interpretable machine learning models, e.g. logical analysis of data (see https://www.mdpi.com/2073-8994/14/3/600 ), decision trees etc.

Answer: Artificial intelligence technologies (i.e. deep learning, machine learning, and rules-based) are mentioned to show interpretability as a common limitation. The available contributions in the interpretability don’t cover all the running possibilities of the AI algorithms and the solutions are in under development to achieve a comprehensive model of interpretability.

The abstract was updated to show some details about our contribution

Question: The introduction is senseless. The authors do not describe the main idea of their interpretable model or method. What is the difference between their contribution and known results?

Answer: An update was done for the introduction section to introduce our contribution. Moreover, the motivation section was added.

Question: In Section 2, the authors list only two approaches to building interpretable models with "solid theoretical backgrounds". However, modern literature contains much more approaches.

Answer: As mentioned, our model is designed is based on implementing the statistics and the probabilities and the mentioned approaches in the background section are based on the probabilities concept too. These studies were analyzed to achieve the state of the art in the field where we can place our contribution. However, the other interpretability approaches are mentioned in section (3).

Question: Thus, the bibliography overview is far from complete, and not systematized.

Answer: The bibliography overview is categorized according to the interpretability models of the AI techniques (i.e. deep learning models, machine-learning models, and rules-based models) to facilitate the reader's understanding of the evolution in the interpretability of each AI technique. On the other hand, we aim to promote our contribution as a comprehensive interpretability solution that is applied to all AI algorithms of all techniques.

Question: The proposed algorithms are given without needed details, the authors do not use mathematical notation usual for the works on AI.

Answer: Please note that all the interpretability techniques and algorithms are mathematically and statistically coined face, therefore there are no specific notations generally linked to the interpretability techniques. In our work, as it is based on statistics and probability concepts, we use the related notations e.g., weight, average, probability, and total.

Question: The most important flaw of this work is that there is no comparison of the results with the known interpretable AI methods which can be used for the same medical problems.

Answer: The main objective of our paper is to design a new model which added to the state of the art of interpretability techniques as a contribution that is based on statistics and probability concepts to produce accurate interpretations for the predictions of the AI algorithms. However, comparing the accuracy of various interpretability techniques would be included in an upcoming survey.

Question: Thus, it is impossible to estimate the importance of the results of this paper.

Answer: Kindly note that interpreting the predictions of the AI models as a whole and the neural network models exclusively is promising. However, our model represents an evolution in giving an accurate interpretation in terms of valuable process and prediction. A possible next step is to reduce the complexity of the interpretable rules-based algorithms that offsets their interpretability. Another future work is to investigate how global interpretability can aid in ensuring fairness in artificial intelligence algorithms

Round 2

Reviewer 1 Report

I can't change my opinion about the manuscript, because the modifications done by the Authors since the first review are not the 'major improvements'. Only a few parts of the text have been added, which cat't be considered as the formal problem statement, description of the need and the methodology of image processing (to obtain data presented in Table 8, page #8), etc. etc.

Still, I could not built the same system, based on the information presented in the manuscript. And, in my opinion, such an ability (to disseminate innovative concepts, methodologies, and algorithms) is one of the most important role of research papers.

Author Response

Thank you for your valuable comments. Kindly note that major improvements were added to almost all sections in the manuscript. In addition to adding new sections.

Please click on the link below to view the formatted manuscript without markups.

https://www.dropbox.com/s/c2weq1iznh4ooen/Final%20version%20explainabiliteHealthcareModel2022.docx?dl=0

Kind regards

Reviewer 2 Report

"The basic principle of the new model is in finding the relative weights
of the attributes which represent their relative importance in determining the prediction and the
probability of having the disease. The attributes are either the symptoms of the patient or the
characteristics of the affected parts of the organ as shown in medical image
".

This principle is implemented in the other methods, e.g. SVM. What is the most important feature of the proposed approach?

"Answer: The bibliography overview is categorized according to the interpretability models of the AI techniques (i.e. deep learning models, machine-learning models, and rules-based models) to facilitate the reader's understanding of the evolution in the interpretability of each AI technique. On the other hand, we aim to promote our contribution as a comprehensive interpretability solution that is applied to all AI algorithms of all techniques"

The interpretability is one of the most important requirement in medical applications. The bibliography is very wide. However, the bibliography overview is rather short. The authors aim at buildiang an all-purpose solution "applied to all AI algorithms and all techniques" but they analyze only few algorithms and techniques, mostly of Chinese authorship.

"Answer: Please note that all the interpretability techniques and algorithms are mathematically and statistically coined face, therefore there are no specific notations generally linked to the interpretability techniques. In our work, as it is based on statistics and probability concepts, we use the related notations e.g., weight, average, probability, and total."

That's true. However, the results must be reproducible. That's why the authors must use one of commonly used algorithm notations. The  algorithm descriptions was improved in the last version.

"Answer: The main objective of our paper is to design a new model which added to the state of the art of interpretability techniques as a contribution that is based on statistics and probability concepts to produce accurate interpretations for the predictions of the AI algorithms. However, comparing the accuracy of various interpretability techniques would be included in an upcoming survey."

To add something to the "the state of the art of interpretability techniques ", the authors must demonstrate the the competitiveness of new technique, at least in comparison with the most common of other methods.

Author Response

Thank you for your valuable comments. Kindly note that major improvements were added to almost all sections in the manuscript.

Please click on the link below to view the formatted manuscript without markups.

https://www.dropbox.com/s/c2weq1iznh4ooen/Final%20version%20explainabiliteHealthcareModel2022.docx?dl=0

Kind regards